# Diosgenin Loaded Polymeric Nanoparticles with Potential Anticancer Efficacy

**DOI:** 10.3390/biom10121679

**Published:** 2020-12-16

**Authors:** Nikita Sharma, Monisha Singhal, R. Mankamna Kumari, Nidhi Gupta, Romila Manchanda, Asad Syed, Ali H. Bahkali, Surendra Nimesh

**Affiliations:** 1Department of Biotechnology, School of Life Sciences, Central University of Rajasthan, Ajmer 305817, India; 2015phdbt01@curaj.ac.in (N.S.); 2014phdbt005@curaj.ac.in (R.M.K.); 2Department of Biotechnology, IIS (Deemed to be University), Jaipur 302020, India; monishasinghal26424@iisuniv.ac.in (M.S.); nidhi.gupta@iisuniv.ac.in (N.G.); 3School of Basic and Applied Sciences, K.R. Mangalam University, Gurugram 122103, India; romila_m@hotmail.com; 4Department of Botany and Microbiology, College of Science, King Saud University, P.O. Box 2455, Riyadh 11451, Saudi Arabia; assyed@ksu.edu.sa (A.S.); abahkali@ksu.edu.sa (A.H.B.)

**Keywords:** PGMD nanoparticles, diosgenin, Box-Behnken design, anticancer

## Abstract

This study aims to determine the anticancer efficacy of diosgenin encapsulated poly-glycerol malate co-dodecanedioate (PGMD) nanoparticles. Diosgenin loaded PGMD nanoparticles (variants 7:3 and 6:4) were synthesized by the nanoprecipitation method. The synthesis of PGMD nanoparticles was systematically optimized employing the Box-Behnken design and taking into account the influence of various independent variables such as concentrations of each PGMD, diosgenin and PF-68 on the responses such as size and PDI of the particles. Mathematical modeling was done using the Quadratic second order modeling method and response surface analysis was undertaken to elucidate the factor-response relationship. The obtained size of PGMD 7:3 and PGMD 6:4 nanoparticles were 133.6 nm and 121.4 nm, respectively, as measured through dynamic light scattering (DLS). The entrapment efficiency was in the range of 77–83%. The in vitro drug release studies showed diffusion and dissolution controlled drug release pattern following Korsmeyer–Peppas kinetic model. Furthermore, in vitro morphological and cytotoxic studies were performed to evaluate the toxicity of synthesized drug loaded nanoparticles in model cell lines. The IC_50_ after 48 h was observed to be 27.14 µM, 15.15 µM and 13.91 µM for free diosgenin, PGMD 7:3 and PGMD 6:4 nanoparticles, respectively, when administered in A549 lung carcinoma cell lines.

## 1. Introduction

Cancer remains one of the most devastating diseases afflicting the world and contributes to about 10 million new individuals every year. It is the second leading cause of death after cardiovascular diseases, making it a serious threat to human society. According to the WHO report, one in every six deaths in the world is caused due to cancer [1]. Various therapeutic strategies are being practiced as a first line of treatment including surgery, radiotherapy, chemotherapy, combination therapy, and laser therapy. Chemotherapy is generally given as adjuvant and neoadjuvant therapies, depending upon the type and stage of cancer. These chemotherapeutic approaches are often associated with toxicities including long-term cardiotoxicity, neurotoxicity, neuropathy, bone marrow suppression, chronic liver damage, gastrointestinal lesions, infertility and many others [2]. Moreover, 90% of cancer deaths are due to the development of resistance as the treatment proceeds [3,4,5]. With the increase in incidence rates and poor prognosis, there remains a continuous challenge for the proper management and treatment of cancer. To overcome the drawbacks of traditional strategies, researchers are searching for some alternatives to improve the efficiency and lower toxicities.

Since ancient times, natural plant-derived compounds have been used as remedies against various diseased states. Numerous studies have demonstrated that these compounds have immense potential to reduce cell growth, inhibit metastasis and angiogenesis, and induce cell apoptosis [6]. Recently, steroidal saponins and their derivatives have been shown to possess tremendous anticancer activity [7]. Diosgenin is a phytosteroidal saponin present abundantly in a variety of plants including *Trigonella foenum graecum*, *Dioscorea villosa*, *Costus speciosus* and many more. It has been shown to induce apoptosis in different cancer cells, by suppressing various pro-inflammatory and pro-survival cell signaling cascades [8]. Many pre-clinical studies have shown the anticancer effect of diosgenin in various cancer cells such as breast, lung, colon, hepatocellular carcinoma, prostate, squamous carcinoma and others [9]. In one of the studies, the anticancer effect of diosgenin was investigated against HepG2, hepatocellular cell lines. The results showed apoptotic and anticancer potential on cancer cells by significantly inducing the death receptors (DR4) and caspase-3 levels [7]. It was shown to inhibit cell proliferation by induction of apoptosis and autophagic activity through the inhibition of PI3/Akt/mTOR signaling pathways in DU145, prostate cancer cells [10]. In another study, the anti-metastatic potential of diosgenin was evaluated through transwell assay in MDA-MB-231 breast cancer cell lines. It was observed that it significantly inhibited the migration of cells when administered with diosgenin. It suppressed actin polymerization, Vav2 phosphorylation, which might be the reason for its anti-metastatic activity in cancer cells [11]. Many studies suggested that the diosgenin works by altering multiple cell signaling pathways, which are involved in significantly inhibiting cell proliferation, migration, angiogenesis, differentiation and enhancing the apoptotic activities [12]. Moreover, the anticancer effect of diosgenin has been shown to be associated through the induction of p53 expression, G1/M cell cycle arrest, immune-modulation and caspase-3 activity, and stimulating the STAT3 signaling pathways [13]. It also suppressed the cell proliferation of cholangiocarcinoma cells by inhibiting the expression of cyclin B1 and enhancing the cell cycle inhibitor p21 levels. This resulted in the cell cycle arrest at G1/M phase. It induced apoptosis through GSK3beta/beta-catenin pathway by following the increase in expression of cytochrome-c, Bax/Bcl-2 ratio, cleaved caspase-3, and cleaved PARP1 [14].

Diosgenin and other plant-based compounds are associated with low toxicities compared with pharmaceutical drugs, which facilitates their use in curing various diseases. Despite their low side-effects, their pharmaceutical use has been limited owing to their poor water solubility (0.02 mg/L) and permeability, low stability and pharmacological bioavailability [9,13]. This further decreases their therapeutic efficacy against cancer. Therefore, an efficient therapeutic approach is required that enhances the availability of the drug in the biological system, thereby improving the anticancer efficacy of the molecule.

Nanotechnology-based methods provide an ideal vehicle for the delivery of pharmaceutical materials in the body. Modulating their size and its surface chemistry, one can tailor the functionalities of the nanocarrier according to their biomedical application. Some of the nanocarriers used for the delivery includes the liposomes, dendrimers, fullerenes, nanorods, and polymeric nanoparticles. Among these, polymer nanocarriers have been extensively used for the targeted delivery of the anticancer drug [15]. Polyester-based drug delivery systems are known to possess a good shelf life, adequate physicochemical properties, and well-defined degradation products. Poly(lactic-co-glycolic) acid (PLGA) is one of the most commonly used polyester-based polymers for delivering the drugs to the targeted site. However, due to its high glass transition (Tg) temperature (45 °C–50 °C), it is not sensitive to external heat (about 43 °C). Furthermore, the unmodified PLGA is hydrophobic in nature, and facilitates the encapsulation of hydrophobic drugs only. This limits its use [16]. In this study, we explored the drug delivery efficiency of a polymer known as poly-glycerol malate co-dodecanedioate (PGMD). As shown in the literature, the polymer was synthesized using a thermal condensation technique of mixing glycol, malic acid, and dodecanedioic acid (DDA). The basic idea for the synthesis of the novel polymer was derived from the work of Migneco et al. [17]. The authors synthesized poly-glycerol-dodecanoate (PGD) as an attractive polymer for biomedical applications. The monomers, glycerol, and dodecanedioic acid react to form esteric bonds that are hydrolysable. Furthermore, these monomers, i.e., glycerol, act as a precursor molecule for the synthesis of triacylglycerols and phospholipids in liver and adipose tissue. They also get converted to glyceraldehyde-3-phosphate and enter the gluconeogenesis and glycolysis pathway. On the other hand, dodecanedioic acid plays an important part in the β-oxidation pathway [18]. However, the synthesized polymer was biodegradable and biocompatible, and it was more desirable for the development of scaffolds (tissue engineering) and surgical implants. Thus, incorporation of malic acid in PGD adds hydrophilicity and change in glass transition temperature (Tg) that becomes more compatible for the entrapment of hydrophobic drugs and proteins/DNA. Consequently, due to the chemical properties such as hydrophilicity, the glass transition temperature of the polymer can be modified with the change in the ratio of malic acid to DDA during the synthesis process. The flexibility and ease of modification of the polymer, allows control of the encapsulation and release profile of the respective drug. It exhibits beneficial properties such as biocompatibility and biodegradability as it can be easily broken into by-products of glycol, malic acid and DDA [16,19,20].

Recently, Erdagi et al. developed a co-delivery system in which diosgenin was chemically conjugated with polycaprolactone (PCL) constituting a hydrophobic core, and methoxy polyethylene glycol (MPEG) as outer shell. Results revealed the formation of uniform spherical nanoparticles as measured through TEM and DLS. Furthermore, the in vitro anticancer activity was evaluated against human fibroblast (L929), breast carcinoma (MCF-7), and osteosarcoma (SAOS-2) cells. It was observed that the formulated nanoparticles showed lower IC_50_ value compared to the free drug [21]. Similarly, a pH sensitive diosgenin nanocarrier platform was developed to enhance the efficiency of doxorubicin for their synergistic delivery against cutaneous melanoma. The results dictate inhibition of metastasis and apoptosis through the mitochondria associated inhibition pathway. The in vivo results showed that the nanoformulation inhibited tumor metastasis and increased apoptosis by employing synergistic effect at the targeted site [22]. Moreover, diosgenin niosomes were developed with spherical morphology and uniform size distribution. The results demonstrated an enhanced anticancer effect in HepG2 cells by improving the cell toxicity from 38.75% for free diosgenin to 71.68% for diosgenin loaded niosomes [9].

The rationale of this study is to explore the efficacy of PGMD polymeric nanoparticles to deliver diosgenin to the cancer cells and to evaluate their anticancer efficiency. The present study focuses on the synthesis and characterization of diosgenin encapsulated PGMD nanoformulations with varying polymer compositions of 7:3 and 6:4 (DDA: malic acid ratio). Synthesis of polymeric nanoparticles is dictated by several factors, hence the optimization of nanoparticles synthesis using Design of Experiments (DoE) has been undertaken in the present work. The preparation of PGMD nanoparticles was systematically optimized employing Box-Behnken design taking into account the influence of various independent variables (factors) such as concentrations of each PGMD, diosgenin and Pluronic F-68 (PF-68) on the responses such as size and polydispersity index (PDI) of the particles. Encapsulation efficiency and drug release kinetics of the formulated nanoparticles were also studied. Microscopic examination was performed using DAPI staining, which showed the induction of apoptotic processes in the treated cells. Furthermore, the anticancer efficiency of the particles was investigated against A549 lung carcinoma cell lines. The efficiency and IC_50_ was determined through Alamar Blue assay.

## 2. Material and Methods

### 2.1. Materials

Malic acid, Diosgenin, Pluronic F68, 1, 12-Dodecanedioic acid (DDA), Pluronic 127 were purchased from Sigma Aldrich. Acetone was obtained from Sisco Research Laboratories Pvt. Ltd. (Mumbai, India). Perchloric acid was purchased from Central Drug House (CDH) (Mumbai, India). For animal cell culture studies, Phosphate buffered saline, Alamar Blue and DAPI or 4′,6-diamidino-2-phenylindole, sodium bicarbonate were procured from HiMedia (Mumbai, India). Dulbecco’s modified Eagle’s medium (DMEM), Fetal Bovine Serum (FBS), Trypsin-EDTA (0.25%) were purchased from Gibco Life Technologies (Carlsbad, CA, USA). MilliQ water was used in the preparation of solutions during the study. A549, human adenocarcinoma cell lines were purchased from NCCS, Pune, India.

### 2.2. Polymer and Nanoparticles Synthesis

The PGMD polymers were synthesized with slight modifications of the protocol as suggested by Migneco et al. [17]. Two variants of the polymer were synthesized by changing the molar ratio of DDA and malic acid (7:3 and 6:4). Briefly, DDA and malic acid were mixed at definite ratios in the presence of glycerol and heated up to 120 °C for 48 h [19].

Nanoparticles were prepared through nanoprecipitation method or solvent displacement technique. Briefly, 5 mg PGMD polymer was dissolved in 5 mL acetone containing 0.5 mg of diosgenin through pipetting or vortexing. The drug-polymer solution was added dropwise into 5 mL of 0.1% PF-68 in a round bottom flask kept on a magnetic stirrer (UC152, Stuart hotplate stirrer, Biocote, St. Neots, UK). The flask was kept overnight on continuous stirring to remove the solvent. The particles were pelleted out at 12,000 rpm for 20 min at 4 °C (Hereaus Fresco 17, Thermo Scientific, Waltham, MA, USA). The particles were lyophilized and stored at 4 °C until further use.

### 2.3. In Silico Optimization of Parameters Using Box-Behnken Design (BBD)

All the parameters optimization was done on Design Expert using Box-Behnken design (BBD) version 12 (Stat-Ease Inc., Minneapolis, MN, USA). Different parameters that control the synthesis of diosgenin loaded polymeric nanoparticles were optimized, that includes concentration of PGMD (mg), Diosgenin (mg) and PF-68 (%) (Table 1). These factors were used as independent variables that yield particle size and PDI (dependent variable) as responses. The optimization of these independent variables was done at two levels, i.e., low to high levels. These conditions suggested 17 experimental trials as given in Table 2. The obtained data were then analyzed by using the Quadratic second order modeling method to determine factors like ANOVA, coefficient of correlation (r^2^), and adjusted/predicted r^2^. The numerical and graphical techniques were used to identify the optimized condition recommended for the synthesis process.

### 2.4. Physicochemical Characterisation

The prepared nanoformulations were characterized for their size, zeta potential and polydispersity in the solution. For this, the pellet obtained after centrifugation was washed with distilled water two–three times to ensure complete removal of unencapsulated or free diosgenin. Then, the pellet was suspended in double distilled water for characterization studies. The particles were analyzed through dynamic light scattering technique as measured by Zetasizer (Zetasizer Nano Series, Malvern PanAnalytical, Malvern, UK). The drug encapsulation efficiency and in vitro drug release of diosgenin from PGMD nanoparticles (7:3, 6:4) were determined through UV-vis spectrophotometer (Evolution 201, UV-vis spectrophotometer, Thermo Scientific, Waltham, MA, USA).

### 2.5. Drug Loading and Encapsulation Efficiency

The drug loading (%DL) and encapsulation efficiency (%EE) of PGMD-diosgenin nanoparticles (PGMD-DG NPs) were estimated spectrophotometrically. Initially, 5 mg freeze-dried nanoparticles were dissolved in DMSO (1 mg/mL) and incubated for 2 h in an incubator shaker so that the complete drug was extracted from the nanoparticles. Perchloric acid, a color developer, was added to the resulting solution. The absorbance was recorded at 410 nm wavelength. The amount of drug was calculated through the standard plot of diosgenin with varied known concentrations. All the experiments were performed in triplicates. A standard graph of diosgenin was plotted over a considerable concentration of 0.0625 mg/mL–5 mg/mL and was found to be linear with r^2^ value of 0.99.
(1)$$\%DL=\frac{\left[Weight\;of\;diosgenin\;in\;NPs\right]}{\left[Weight\;of\;NPs\right]}\;\times \;100$$

The %EE was enumerated indirectly through centrifugation. The synthesized nanoparticles were pelleted by centrifugation at 12,000 rpm for 20 min at 4 °C. The supernatant containing the free diosgenin was estimated through UV vis-spectrophotometer. The amount of drug was determined through the standard curve of diosgenin at a particular wavelength.
(2)$$\%EE=\frac{\left[Amount\;of\;diosgenin\;encapsulated\;in\;the\;nanoparticles\right]}{\left[Total\;amount\;of\;diosgenin\;used\;during\;preparation\right]}\;\times \;100$$

### 2.6. In Vitro Drug Release Kinetics

Determination of the amount of drug released from the nanoparticles was done through spectroscopic absorption. For this, the drug loaded nanoparticles were dispersed in phosphate buffer saline (PBS) of varying pH 5.4 and 7.6 and were incubated in a shaker at 37 °C, for different time periods of 1, 2, 3, 4, 5, 10, 24 and 48 h. At different time intervals, the aliquots were taken out and centrifuged at 12,000 rpm for 20 min. The supernatant contains released diosgenin, which was further measured through a spectrophotometer. Perchloric acid was added as a color developer to the supernatant and further absorbance was recorded at 410 nm wavelength. To investigate the release mechanism and kinetics of diosgenin from the nanoparticles, the results were fit in various kinetic models. The model following the best fit will represent the release pattern of the drug from the nanoparticles.

### 2.7. Apoptosis Analysis through DAPI Staining

Alterations in nuclear morphology such as chromatin condensation, nuclear fragmentation, and cellular budding occurs in the apoptotic process. These can be determined by using fluorescence microscopy and fluorescent dyes which stain the cellular nucleus. The apoptotic analysis was performed by using cell-permeable nuclear stain, DAPI. Briefly, the A549 cells were seeded in a 96-well-plate at a cell density of 8 × 10^3^ cells per well. The plate was kept in an incubator at 37 °C, 5% CO_2_ overnight to allow adherence of cells to the plate. After overnight incubation, the cells were administered with free diosgenin, PGMD-DG 7:3 NPs, PGMD-DG 6:4 NPs. The cells in DMEM media without treatment were considered as control. All the experiments were performed in triplicates. Following 24 h incubation, the wells were washed with PBS twice to ensure complete removal of free drug and nanoparticles. These cells were fixed with 4% formaldehyde for 15 min, followed by DAPI staining at a 3 nM concentration for 10 min. The images were examined and captured under fluorescence microscope (Leica, DMI 6000B microscope, Wetzlar, Germany).

### 2.8. Acridine Orange/Ethidium Bromide Staining for Apoptotic Analysis

Fluorescence microscopic analysis of apoptosis was done through Acridine orange/Ethidium bromide (AO/EtBr) double staining assay. It is used to monitor nuclear changes and apoptotic body development, which are distinguishing features of apoptosis [23]. For this, A549 cells were seeded at a cell density of 8 × 10^3^ cells per 100 µL in 96-well-plate. After 24 h incubation, the cells were treated with Diosgenin, PGMD-DG NPs 7:3 and PGMD-DG NPs 6:4 for 24 h at 37 °C and 5% CO_2_. The cells were washed thrice with PBS and were stained with AO/EtBr (AO-100µg/mL and EtBr-100 µg/mL at 1:1 ratio). The cells were washed with PBS after 2–3 min staining, and were visualized under fluorescence microscope (Leica, DMI 6000B microscope, Wetzlar, Germany). The images have been quantitated for apoptosis analysis using Image J version 1.52B.

### 2.9. Cytotoxicity Assay on A549 Cells

To assess the efficiency of the formulated nanoparticles, cytotoxicity assay was performed using Alamar Blue assay. As discussed earlier, cells were seeded at cell density of 8 × 10^3^ cells/100 µL. These were allowed to incubate at 37 °C, 5% CO_2_ overnight in an incubator. The cells were treated with different concentrations of diosgenin nanoparticles and diosgenin alone with proper control (cells with media containing DMEM and FBS). The cells were kept in a CO_2_ incubator for 24 and 48 h time period. There was 10% resazurin (0.15 mg/mL in PBS) added to each well and incubated for 2–4 h in an incubator. The metabolic activity of cells as displayed by conversion of the blue color to the pink one after treatment was measured using Elisa plate reader (Thermo Multiskan GO, Thermo Scientific, Vantaa, Finland). The difference in absorbance was observed at 570 nm and 600 nm, respectively.

### 2.10. Statistical Analysis

The results were calculated from 3 independent experiments. Error bars were showed as mean ± standard deviation (SD) and statistical analysis was performed through GraphPad prism software (version 6.01, GraphPad Software Inc., San Diego, CA, USA). Analysis of variance (ANOVA) was employed to compare the treated with the control groups. The data with *p* value < 0.05 were considered statistically significant.

## 3. Results and Discussion

This study was undertaken to explore the potential of PGMD nanoparticles to deliver diosgenin to the lung cancer cells. Herein, two different variants of the polymer, i.e., PGMD 7:3 and PGMD 6:4, have been employed for the preparation of nanoparticles and treatment of the lung cancer cells. Polyester poly (glycerol-dodecanoate) (PGD) were earlier synthesized using glycerol and dodecanedioic acid (DDA) through ester bonds by Migneco et al.; they were reported to exhibit good mechanical and biological properties [17]. Although PGD is biodegradable and biocompatible, it is highly hydrophobic and has low glass transition temperature (Tg) (32 °C), which masks its application in nanoformulations and drug delivery [18,24]. Addition of malic acid to PGD polymer remarkably influenced the Tg (42.2 °C) and hydrophilicity of the polymer, different variants were prepared with the ratios of 7:3 and 6:4 for DDA and malic acid, respectively [16,19]. The MW of the PGMD polymer was observed to be 3000 Da as measured by GPC column and Evaporative Light Scattering Detector (ELSD) and have been reported in one of our earlier publications. Furthermore, the FT-IR studies of PGMD 7:3 polymer revealed the presence of a C=O stretch at 1735 cm^−1^, which is typical of ester bonds and relevant chemical shift peaks were observed in ^1^HNMR for PGMD (Appendix A) [16].

### 3.1. Optimization of Diosgenin Loaded Polymeric Nanoparticles

The systematic optimization of parameters using Box-Behnken design (BBD) suggested a total of 17 experimental trials. Once the data were generated, quadratic polynomial modeling and various statistical analyses were performed. As a result of quadratic polynomial modeling, equations for both the responses were obtained showing interaction and curvature effects for both the responses. Furthermore, the model diagnostic plots for both the responses are interpreted in Figure 1 implying good fitting of the evidence. In Figure 1A, predicted vs. actual plot shows the studentized or standardized values vs. predicted ones. This graph was quite linear, signifying that experimental values were quite close in proximity with the predicted ones. The perturbation plot aids in comparison among the effects of all three independent variables at a particular point in the given design space. In Design Expert, by default this reference point is set at the midpoint, i.e., 0. Factor A and C with slight curvature depicts that they are sensitive to the response-particle size, which means change in these two factors could affect the response to a certain extent. On the other hand, the flat line of factor B suggests that response remains unaffected. Interaction plot was used to determine the mean effects of one factor (PGMD) with the other selected factor (diosgenin). The red and green mean effects lines of both the factors intersect with each other, which means there is interaction between the two. Similarly in Figure 1B, the graph obtained for the PDI was linear, signifying that some experimental values lying on the linear line have a close association with the predicted ones. The perturbation plot shows curvature for all the three factors, depicting that they are sensitive to the response. Interaction plot also shows intersecting mean effects lines for PGMD and diosgenin, exhibiting an interaction among the two.

### 3.2. Response Surface Method by Box-Behnken Design

In experimental designing, response surface method (RSM) is an approach that combines mathematical as well as statistical techniques used to obtain optimized conditions for the experiment. It scrutinizes the association between different variables taken into consideration for the experiment. In the current work, the BBD approach was used to obtain optimal conditions for synthesizing diosgenin loaded polymeric nanoparticles. Our aim was to investigate the three important factors required for synthesis process viz., concentrations of PGMD (mg), diosgenin (mg) and PF-68 (%). For the synthesis of polymeric nanoparticles, these factors play an influential role in determining the size and polydispersity index. In a study, the effects of process and variables required for synthesis of parenteral paclitaxel-loaded biodegradable polymeric nanoparticles was evaluated [25]. The study suggested that alterations in such variables might affect the size, encapsulation and drug releases efficacies. Herein, the relationship between independent variables and their effects on response variables were determined by 3D response surface plots and 2D contour plots using response surface methodology (Figure 2 and Figure 3).

The association between PGMD and diosgenin concentration is presented in Figure 2A that shows that increase in PGMD concentration with slightly high levels of diosgenin, shows a moderate decline in particle size. At the highest level of diosgenin concentration along with low levels of PGMD concentration, a slight decrease in the trend of nanoparticles is seen. Likewise, a 2-D contour plot shows the curvatures near high diosgenin concentrations and low PGMD concentrations. Hence, it might be assumed that a small size of nanoparticles is directly proportional to low concentrations of PGMD along with a high concentration of diosgenin. The association between PGMD and PF-68 is presented in Figure 2B where at lower PGMD concentration, the size of nanoparticles tends to decrease when there is a slight increase in PF-68 concentration. The 2-D contour plot also depicts the curvatures near lower PGMD concentrations. Here also, lower PGMD concentrations play an important role in the synthesis of small sized nanoparticles along with increase in PF-68 concentrations. In Figure 2C, the association between diosgenin and PF-68 is presented, which shows the maximum number of particles at lower levels of diosgenin along with higher percentage of PF-68. The 2-D contour plot shows maximum number of curvatures near low diosgenin levels with increasing concentrations of PF-68. Hence, it might be assumed that low diosgenin concentrations play a crucial role in determination of nanoparticle size. Similarly, the relationship of PDI with all the factors was generated by response surface methodology shown in Figure 3. The 3-D plot (Figure 3A) for PGMD and diosgenin shows a declining trend in PDI when the concentration of both the factors tends to increase. The association of PGMD and PF-68 concentrations as illustrated in Figure 3B shows the negative impact on PDI when concentration of both factors increases. In Figure 3C, the PDI values tend to increase at higher concentrations of PF-68 but it seems to be lower in mid levels of diosgenin.

### 3.3. Investigation for Optimized Polymeric Nanoparticles

The hunt of optimized parameters for the synthesis of polymeric nanoparticles was done by numerical optimization by BBD. The desired range for both the response variables, i.e., particle size and PDI were maintained. The yellow region in the overlay plot along with the flagged points described the optimized region for particle synthesis. The desirable conditions as depicted by the overlay plot for the synthesis process includes 4.87 mg of PGMD, 0.10 mg of diosgenin with 0.19% of PF-68 which yields polymeric nanoparticles of 104.5 nm having 0.34 PDI (Figure 4). It was observed that the validated values of nanoparticles were quite close to the predicted values.

### 3.4. Physicochemical Characterisation of Nanoparticles

The diosgenin loaded PGMD nanoparticles were prepared through nanoprecipitation technique (also known as solvent displacement method). This approach of synthesizing nanoparticles is based on various factors including polymer and drug concentration, molecular weight of the preformed polymer, aqueous organic phase volume, stirring speed, stirring time, surfactant concentration, reaction temperature and types of solvent. These parameters control the size and the amount of drug encapsulation in the formed nanoparticles [26]. This method offers several advantages over other synthetic approaches owing to its simplicity, cost, speed, reproducibility and as it requires less energy consumption [27]. During synthesis, polymer was dissolved in acetone, a partially miscible organic solvent. This solution was then added dropwise to 0.1% PF-68 aqueous solution, under continuous stirring. To remove the organic solvent from the solution, it was allowed to stir overnight. Studies have shown that every droplet is responsible for generating numerous nanoparticles following interfacial phenomenon. The formation of nanoparticles could be explained through diffusion stranding mechanism. As the solvent diffuses into the aqueous solution, the interfacial tension between liquid-liquid surface increases, resulting in the formation of smaller and stable nanoparticles. The presence of surfactant stabilizes the nanoparticle formation, and avoids the coalescence and aggregation of particles during the synthesis process [28].

An effective nanocarrier-mediated drug delivery necessarily requires an efficient cellular uptake and permeability of nanoparticles into the tumor. This depends on certain important parameters including size, shape and surface charge density [29]. In this study, the synthesized particles were subjected to DLS and zeta analysis, for the measurement of size, surface charge and polydispersity index of the nanoparticles. Size of the void and drug conjugated polymeric nanoparticles were found to be 111.6 nm, 133.6 nm at 7:3 ratio with polydispersity index (PDI) of 0.111, 0.152 and 114.9 nm, 121.4 nm at 6:4 ratio with PDI of 0.137, 0.270, respectively, with a homogeneous size distribution (Figure 5) (Appendix A). It was evident from the results that size of the nanoparticle increases with the concentration of the drug. This could be attributed to the increased viscous dispersed phase of the drug and polymer solution resulting in the formation of larger particle size. Nanoparticles with size < 200 nm are able to escape the reticuloendothelial system of the body, hence their chances of being eliminated from plasma is low [16]. Thus, the smaller size of these nanoparticles can facilitate an improved cellular uptake in cancer cells [30]. Moreover, the zeta potential of the PGMD-DG 7:3 NPs and PGMD-DG 6:4 NPs were found to be −20.7mV and −24.5mV. This indicated the colloidal stability of the nanoparticles and a lower tendency to aggregate. The presence of carboxylic groups confers the negative charge on the surface of nanoparticles [31]. Similar trends were observed when methoxy poly(ethylene glycol) (MPEG) conjugated diosgenin nanoparticles were prepared [21].

### 3.5. Percentage Drug Loading and Encapsulation Efficiency

The drug loading and encapsulation studies are substantial to improve the potency of therapeutic drugs and further evaluate their application at clinical level. Concentration of drug and polymer, surfactant concentration, and reaction temperature influence the loading and encapsulation efficiency in the nanoparticles. In the present study, %DL and %EE were evaluated through an indirect method using UV-vis spectrophotometer. The absorbance of diosgenin was recorded at 410 nm wavelength. Initially, the polymer drug ratio was varied from 1:1, 5:1, 10:1, 20:1 (PGMD: DG) in the synthesis process. The maximum encapsulation and loading efficiency was observed at 10:1, and this ratio was used for further experiments. The reason for high %EE and %DL at 10:1 could be attributed to the increased hydrophobic interaction between the chemical moieties of PGMD with diosgenin. In addition, the increase in polymeric concentration results in the increase in viscosity of the solution. This resists the diffusion of the drug from the organic solvent to the aqueous solution. Thus, entrapping more drug in the polymeric nanoparticles as indicated in the previous literature [32,33]. Erdagi et al. prepared diosgenin conjugated poly(ε-caprolactone) as a co-delivery system through a solvent evaporation method. The authors reported %EE (60–85%) and %DL (10–15%) for imatinib with good colloidal stability [21]. On further increasing the concentration, large size nanoparticles were formed. Therefore, 10:1 ratio was chosen for optimum size and encapsulation of drug. The obtained results revealed no significant difference in the encapsulation efficiency between both the variants of PGMD nanoparticles. The results as shown (Table 3) revealed the %EE and %DL of PGMD-DG 7:3, 6:4 were 83.34%, 77.16% and 12.68%, 10.95%, respectively. The difference in encapsulation between the 7:3 and 6:4 variant could be due to the change in concentration of DDA, which corresponds to the hydrophobic interaction between the drug and polymer. Thus, resulting in the variation in encapsulation and loading efficiency.

### 3.6. In Vitro Drug Release Analysis

The drug release studies were performed in different pH of phosphate buffer saline under varying time intervals. In vitro drug release studies showed that the release of drug at pH 5.4 was more than at pH 7.6, as shown in Figure 6. The reason could be attributed to the accelerated hydrolysis of the ester bond present in the PGMD polymer. This resulted in a faster degradation of the polymer and higher diosgenin release from the nanoparticle [34]. It has been stated in the literature that tumor sites have lower pH than blood and healthy tissues. Moreover, the results showed an initial burst release during initial 5–8 h of incubation. The burst release for diosgenin could be associated with the release of non-encapsulated or loosely bound drug in the polymer matrix, which was easily accessible on hydration of the nanoparticles [35]. The release was slower for the next 24 h with up to 70% release (for both the PGMD variants, i.e., 7:3, 6:4), due to the hydrophobic interaction of polymer-drug within the nanoparticle. This resists the fast diffusion of the drug from the particle and maintained its sustained release for long hours. In one of our earlier publications, polyethylenimine-modified PLGA nanoparticles also showed maximum release of epirubicin and paclitaxel at pH 5.4 as compared to pH 7.6 [36].

Furthermore, the drug release data were fitted into different kinetic models (such as zero order, first order, Higuchi model and Korsmeyer-Peppas model) to understand the drug release pattern from the polymeric nanoparticles. The best-fitted release model was selected based on the correlation coefficient (r) as enumerated from the linear regression curve. The correlation coefficient showed that diosgenin release from PGMD-DG 7:3 and PGMD-DG 6:4 NPs followed Korsmeyer-Peppas model (r^2^ = 0.931,0.921) as shown in Table 4. This suggests that the drug release followed diffusion and dissolution controlled release pattern from the polymeric matrix [37]. Moreover, the slope exponent value (*n*) for drug release was calculated to further confirm the dissolution and diffusion mechanism of drug release from the polymer matrix. In general, if the exponent value *n* ≤ 0.45, it stipulates to follow Fickian diffusion drug release characterized by shorter relative polymeric relaxation time than the diffusion time of solvent, whereas 0.45 < *n* < 0.89, designates anomalous type transport following non-Fickian drug release, suggesting diffusion and erosion release mechanism. The results showed that the slope exponent *n* value was smaller than 0.45 for both the formulations. This indicates that the formulation followed non-Fickian type drug release. The reason could be attributed to the increase in molecular rearrangement of polymeric chains and drug polymer interaction as the solvent diffuses in the nanoparticles [38,39]. Therefore, this could be useful in cancer therapy for the targeted delivery of the drug and also for the prolonged period of time [40].

### 3.7. DAPI Staining

Programmed cell death is the characteristic feature of apoptotic cells. It is an ATP-dependent process, which includes cell shrinkage, nuclear fragmentation, chromatin condensation, loss of plasma membrane integrity, and induction of cysteine-based aspartate-activated proteases, known as caspases [41]. In the present study, the cytotoxic effect of drug loaded nanoparticles was assessed morphologically by analyzing the cellular function and membrane integrity of the control cells with the treated ones. In this regard, a fluorescence DNA specific DAPI was used in the experiment to demonstrate the chromatin condensation, nuclear fragmentation. As shown in the results (Figure 7), the A549 cells treated with diosgenin only PGMD-DG 7:3, PGMD-DG 6:4 NPs indicated changes in the nuclear chromatin, and formation of apoptotic bodies. This suggested that the treatment with diosgenin and diosgenin-loaded polymeric nanoparticles could induce apoptosis in lung cancer cells. Kim et al. reported that diosgenin induced reactive oxidative species (ROS) production, and this activated apoptosis thorough apoptosis signal regulating kinase-1 (ASK-1). This leads to the stimulation of p-38 MAPK/JNK upstream cascades in HepG2 cancer cells [42]. Furthermore, Moalic et al. demonstrated that diosgenin induced apoptosis in a cyclooxygenase-1 (COX-1) and COX-2 mediated manner by arresting cell cycle at G1 phase, as observed in osteosarcoma cells [43].

### 3.8. Acridine Orange/Ethidium Bromide Staining

The distinctive feature of an apoptotic cell includes cell shrinkage, membrane blebbing, chromatin condensation and nuclear fragmentation. Apoptosis was evaluated by recognizing apoptotic, viable and non-viable cells under fluorescence microscope. Acridine orange is a green fluorescence emitting organic dye that intercalates with the double stranded DNA. Thus, this dye stains both the viable and non-viable cells. Meanwhile, Ethidium bromide stains the non-viable cells and emits red fluorescence by intercalating with the cellular DNA. It was evident from the results that live cells give green color fluorescence and cells during early apoptosis will emit intense green with chromatin condensation (Figure 8). Additionally, the late apoptotic or necrotic body shows orange red nucleus [44]. Similar morphological changes were obtained when diosgenin was administered in HepG2 cell lines.

Quantification of the cells at early and late apoptotic stages was carried out. This was performed by counting more than 300 cells in each experiment, followed by differentiation of apoptotic cells at their various stages [45,46]. The negative control did not show any significant apoptosis. However, it was observed that upon treatment with the nanoformulations the late apoptotic cells increased by 58 and 40%, respectively (Figure 8E), in relation to the cells treated with diosgenin alone (16.6%). This indicates the robust activity of nanoformulations designed against the cancer cells. The results are also supported by the MTT assay performed in the cancer cells. It was observed that diosgenin induced oxidative stress through the production of reactive oxygen species (ROS), further triggering the apoptotic cascades [47]. Mao et al. showed that diosgenin inhibited tumor growth in human cholangiocarcinoma cells (CCA) by arresting the cells at G2/M phase. It induced apoptosis through the increase in the expression of cytochrome-c, cleaved caspase-3 and Bax/Bcl2 ratio. The results further corroborated the nuclear changes and apoptosis, which were visualized microscopically. Our previous study also showed induction of membrane blebbing and chromatin condensation, as signs of early apoptosis when A549 cells were treated with EPI-PTX combination drug and modified PLGA nanoparticles entrapping these drugs [36].

### 3.9. Cytotoxicity Studies

Cell-dependent analysis is widely used for the determination of toxic effect on cell proliferation, which ultimately leads to cell death [48]. In the study, Alamar blue assay was performed for the assessment of drug and their respective nanoparticles against lung cancer cells. It could be inferred from the results (Figure 9) that the diosgenin and PGMD-DG NPs (for both 7:3, 6:4) showed cytotoxic effect in a concentration-dependent manner. It has been shown in our previous studies that void PGMD nanoparticles do not exhibit any toxicity to the mammalian cells [20]. The results revealed diosgenin nanoparticles to be more potent than the free diosgenin. The IC_50_ value obtained for diosgenin only was 31.92 ± 1.23 µM, whereas for the nano-formulations the values were 18.23 ± 3.15 µM (PGMD-DG 7:3) and 16.27 ± 2.79 µM (PGMD-DG 6:4). The IC_50_ value was reduced to 57.11% and 50.97% for PGMD-DG 7:3 and PGMD-DG 6:4, respectively, when administered against lung cancer cells (Figure 9, Table 5). Similar results were observed when Imatinib-diosgenin conjugated PCL-MPEG nanoparticles were introduced for cell toxicity against various cells including L-929, K-562, SAOS-2 and MCF-7 cells. The analysis depicted that the IC_50_ value for ITB loaded-DGN conjugated PCL-MPEG NPs was significantly lower (*p* < 0.05) when compared to the free drug [21]. The enhancement of cytotoxic activity of the drug loaded nanoparticles than the free drug could be attributed to many theories explained in the literature. According to one of them, nanoparticles generate a concentration gradient across the cell surface by getting adsorbed on the cell membrane. This results in the influx of drug through the membrane. Secondly, tumor cells (which exhibit enhanced endocytic activity) could internalize polymeric nanoparticles, allowing the drug to be released inside the cells, thus contributing to an increase in the drug concentration near its site of action [49]. Additionally, it is also possible that this could be the result of a high payload of diosgenin in the nanoparticles. This could result in substantial intracellular delivery even if only a small number of nanoparticles were able to enter the cells [50].

### 3.10. Conclusions

In the present study, diosgenin loaded PGMD (both variants 7:3 and 6:4) nanoparticles were successfully developed through nanoprecipitation technique. The preparation of PGMD nanoparticles was systematically optimized employing Box-Behnken design taking into account the influence of various independent variables such as concentrations of each PGMD, diosgenin and PF-68 on the responses such as size and PDI of the particles. Both the particles were in the size range of 110–130 nm under optimized conditions, without much difference in the colloidal stability. An enhanced %EE (77–84%) and %DL (10–13%) was obtained when analyzed spectrophotometrically. Furthermore, morphological evaluation was performed using DAPI, which suggested that diosgenin induced apoptosis mediated cellular death in cancer cells. The alamar blue assay was employed for the investigation of cytotoxicity. The results showed a dose and time-dependent cytotoxic effect of nanoparticles on cells. In general, PGMD-DG NPs displayed a significant anticancer potential when compared to free drug in cancer cells. Moreover, advanced studies are required to further validate the results in the future. From the results, we can speculate that PGMD-DG NPs could be used as anticancer compounds against cancer therapy in the future.

## Figures and Tables

**Figure 1 biomolecules-10-01679-f001:**
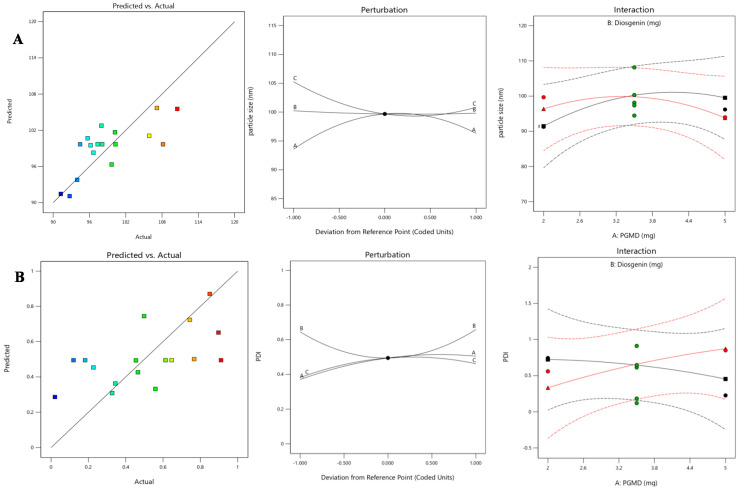
Model diagnostic plots showing predicted vs. actual, perturbation and interaction plots for response variables- (**A**) particle size and (**B**) PDI.

**Figure 2 biomolecules-10-01679-f002:**
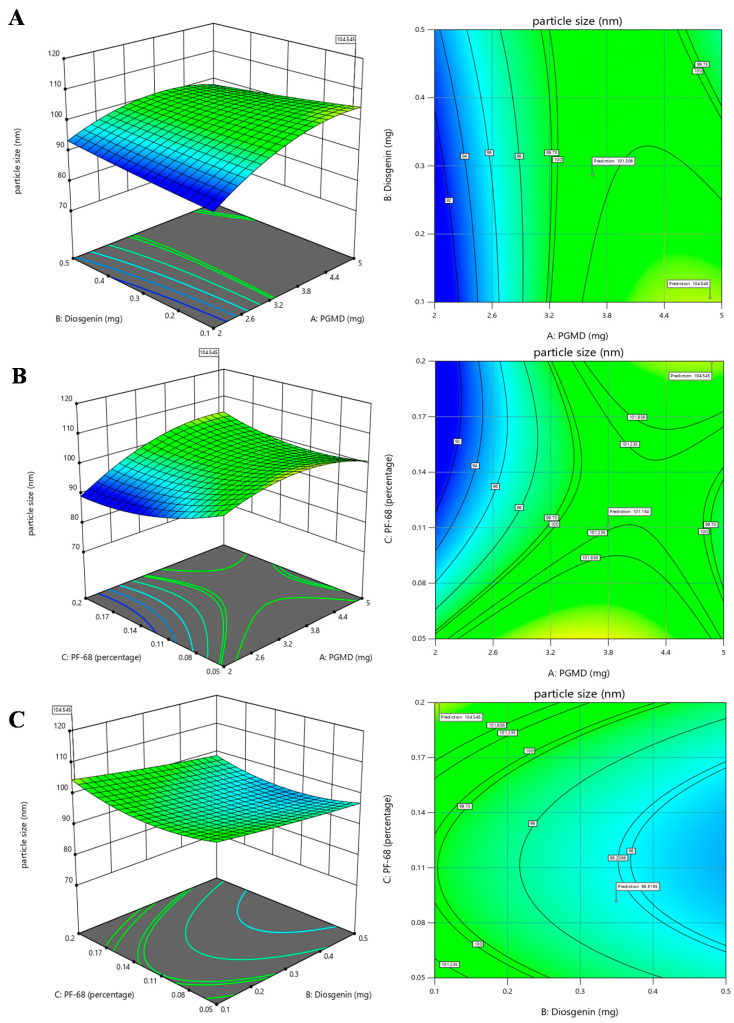
3D response surface plots and 2D contour plots depicting the effects of factors: PGMD, Diosgenin and Pf-68 concentrations on particle size, (**A**) PGMD vs. Diosgenin; (**B**) PGMD vs. PF-68 and (**C**) Diosgenin vs. PF-68.

**Figure 3 biomolecules-10-01679-f003:**
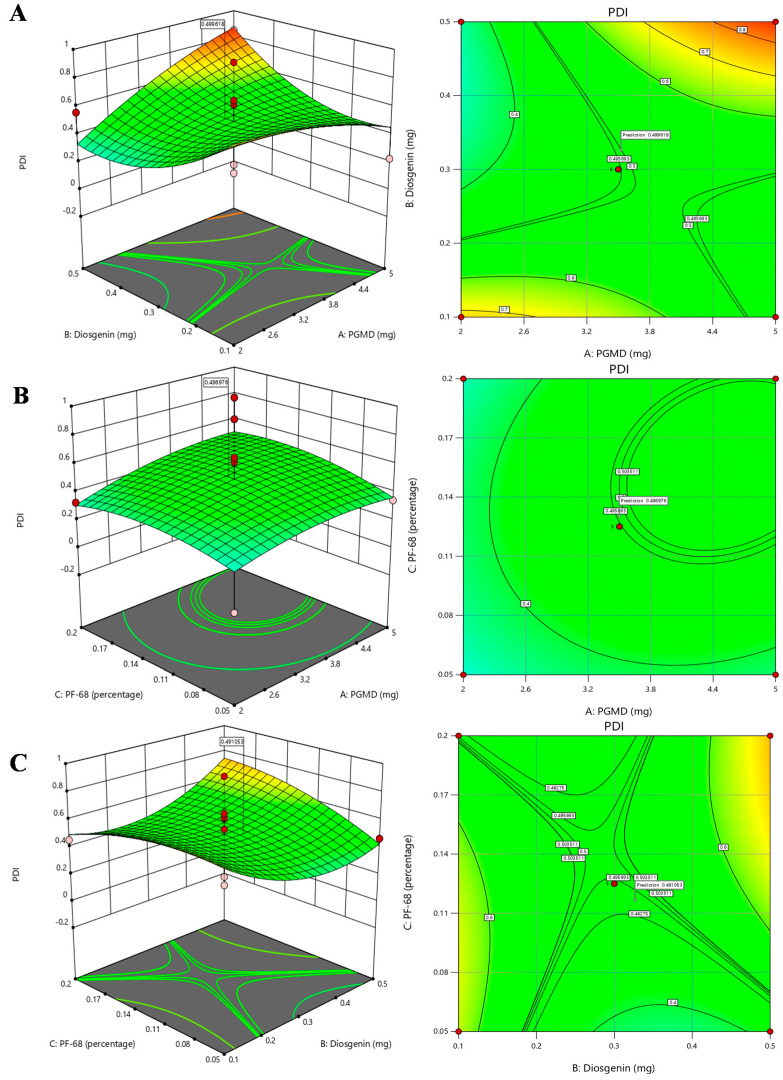
3D response surface plots and 2D contour plots depicting the effects of factors: PGMD, Diosgenin and Pf-68 concentrations on PDI, (**A**) PGMD vs. Diosgenin; (**B**) PGMD vs. PF-68 and (**C**) Diosgenin vs. PF-68.

**Figure 4 biomolecules-10-01679-f004:**
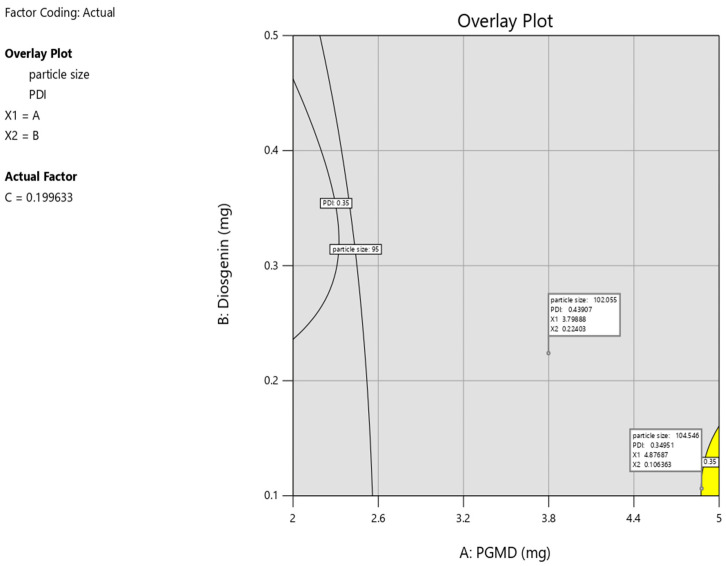
Overlay plot depicts a yellow region as an optimized area and flagged points as the compositions of optimized diosgenin loaded PGMD NPs.

**Figure 5 biomolecules-10-01679-f005:**
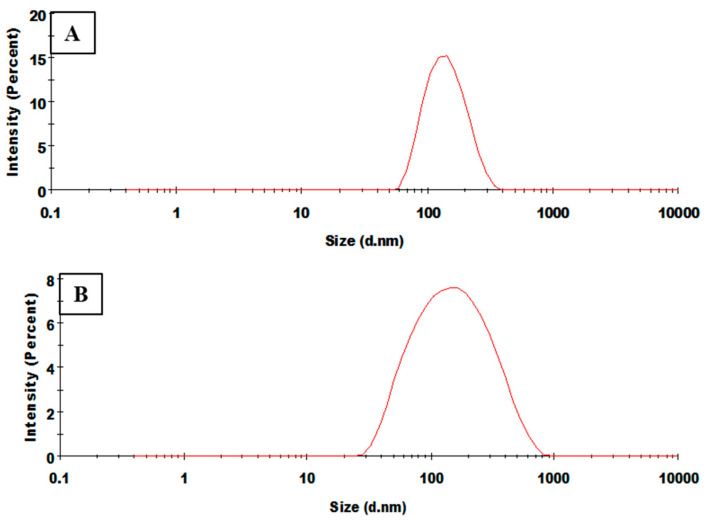
Represents DLS spectrum of (**A**) PGMD-DG 7:3, the particles size is 133.6 nm and (**B**) PGMD-DG 6:4 the particles size is 121.4 nm.

**Figure 6 biomolecules-10-01679-f006:**
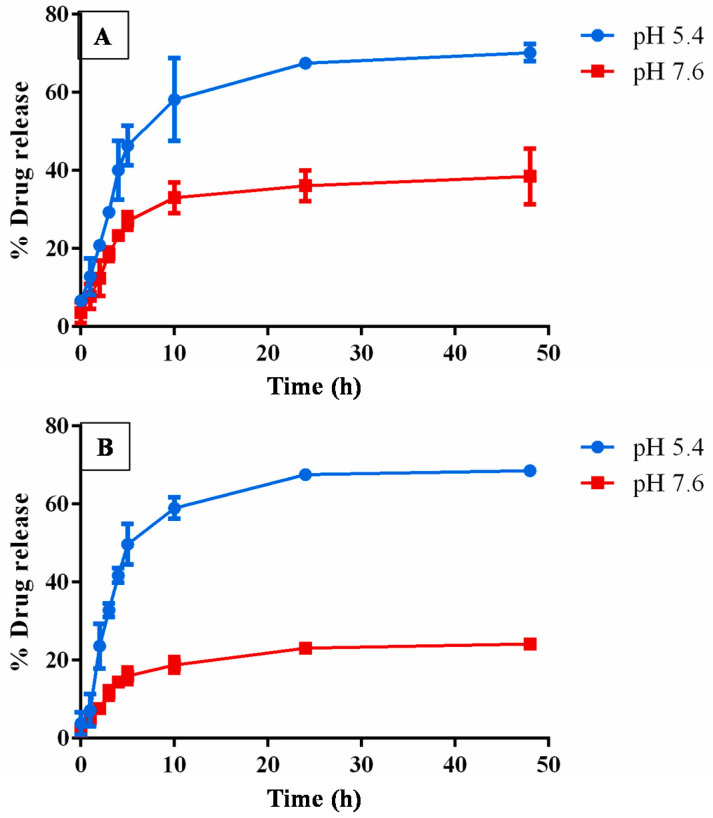
In vitro drug release studies of (**A**) PGMD-DG 7:3, and (**B**) PGMD-DG 6:4 at different pH under varying time intervals.

**Figure 7 biomolecules-10-01679-f007:**
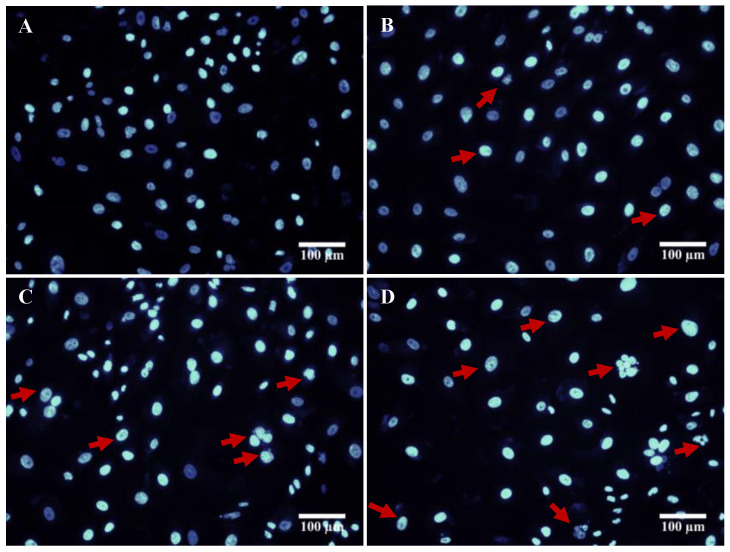
Illustrates DAPI stained fluorescent images of A549 cells (**A**) untreated and treated with (**B**) diosgenin only, (**C**) PGMD-DG 7:3, (**D**) PGMD-DG 6:4.

**Figure 8 biomolecules-10-01679-f008:**
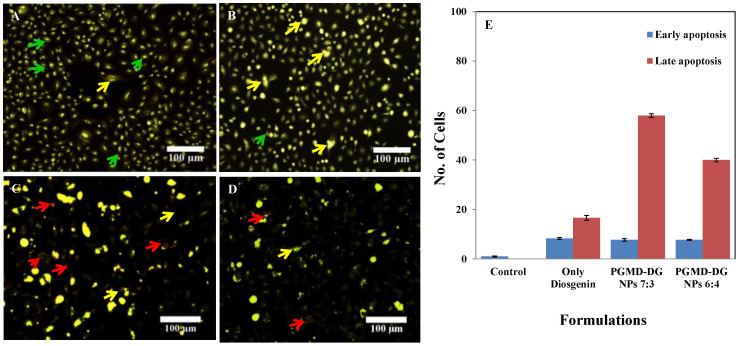
Acridine orange/Ethidium bromide staining in A549 cells. Untreated A549 cells (**A**) showed normal cellular structure (green arrows), early apoptosis (yellow arrows) including chromatin condensation, cell membrane blebbing was observed after treatment with Diosgenin only (**B**), and necrosis (red arrows) with (**C**) PGMD-DG NPs 7:3 and (**D**) PGMD-DG NPs 6:4. (**E**) Quantitative analysis of apoptosis with the treatment of diosgenin PGMD nanoparticles in A549 cells.

**Figure 9 biomolecules-10-01679-f009:**
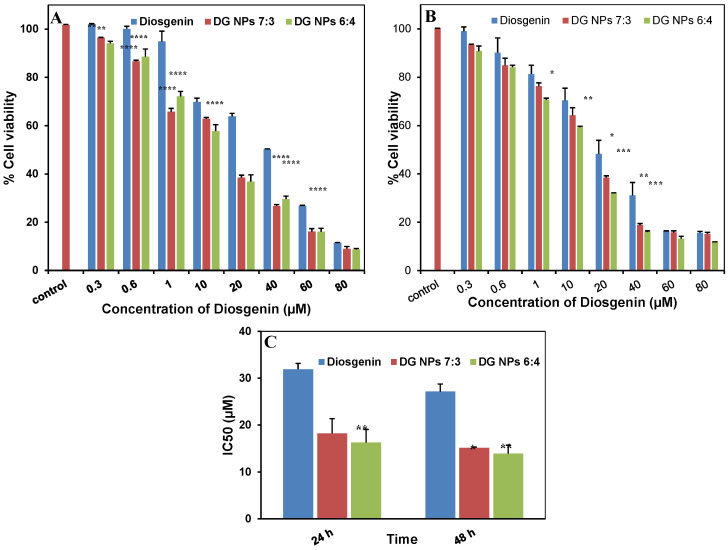
Illustrates the % cell viability of diosgenin only, PGMD-DG 7:3 and PGMD-DG 6:4 NPs in A549 cancer cells at different time points (**A**) 24 h, (**B**) 48 h, and (**C**) represents IC_50_ values for 24 h and 48 h in A549 cancer cells. *, **, ***, **** indicates *p* < 0.05, 0.01, 0.001, 0.0001, respectively.

**Table 1 biomolecules-10-01679-t001:** Experimental factors taken for optimization of parameters for synthesis of diosgenin loaded PGMD nanoparticles.

Factors	Unit	Low	High
PGMD	mg	2	5
Diosgenin	mg	0.1	0.5
PF-68	percentage	0.05	0.2

**Table 2 biomolecules-10-01679-t002:** Experimental design showing trial runs for optimization of parameters for synthesis of diosgenin loaded polymeric nanoparticles.

Run	A:PGMD	B:Diosgenin	C:PF-68	Particle size	PDI
	mg	mg	percentage	nm	
1	3.5	0.3	0.125	97.3579	0.182443
2	5	0.3	0.05	96.6539	0.344201
3	2	0.1	0.125	91.2672	0.743494
4	3.5	0.5	0.05	107.153	0.46536
5	2	0.3	0.2	92.7336	0.327228
6	3.5	0.3	0.125	94.4605	0.910793
7	3.5	0.3	0.125	100.307	0.12029
8	5	0.3	0.2	105.873	0.766865
9	3.5	0.3	0.125	98.1165	0.613436
10	3.5	0.3	0.125	108.157	0.646145
11	2	0.5	0.125	99.6713	0.559112
12	3.5	0.1	0.05	110.518	0.896998
13	3.5	0.1	0.2	100.219	0.454666
14	5	0.1	0.125	96.1892	0.22674
15	5	0.5	0.125	93.9755	0.850414
16	2	0.3	0.05	97.9802	0.0205136
17	3.5	0.5	0.2	95.7049	0.498521

**Table 3 biomolecules-10-01679-t003:** Represents encapsulation and drug loading efficiency of diosgenin loaded PGMD nanoparticles.

Formulations	Polymer: Drug Ratio	Encapsulation Efficiency (%EE)	Loading Content (%DL)
PGMD-DG 7:3 NPs	10:1	83.34 ± 3.67	12.68 ± 1.01
PGMD-DG 6:4 NPs	10:1	77.16 ± 2.61	10.95 ± 0.37

**Table 4 biomolecules-10-01679-t004:** Represents encapsulation and drug loading efficiency of diosgenin loaded PGMD nanoparticles.

Drug Formulation	R^2^ Values			
	Zero Order	First Order	Higuchi Model	Korsmeyer-Peppas Model
PGMD-DG 7:3 NPs	0.724	0.839	0.907	0.931
PGMD-DG 6:4 NPs	0.656	0.685	0.886	0.939

**Table 5 biomolecules-10-01679-t005:** Illustrates the IC_50_ value of free diosgenin and nano-formulations after 24 h and 48 h treatment in A549 cells.

Formulations	IC_50_ (µM)	
	24 h	48 h
Diosgenin only	31.92 ± 1.237	27.14 ± 1.597
PGMD-DG NPs 7:3	18.23 ± 3.159	15.15 ± 0.174
PGMD-DG NPs 6:4	16.27 ± 2.793	13.91 ± 1.803

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
