# Peer review of "Diosgenin Loaded Polymeric Nanoparticles with Potential Anticancer Efficacy"

_biomolecules, 2020, doi:10.3390/biom10121679_

Round 1

Reviewer 1 Report

The study describes an investigation of diosgenin loaded PGMD nanoparticles for potential use in cancer therapy. Though being a very interesting study with not so common polymeric nanoparticles compared to PLGA, the study needs some improvements:

  • the benfit of DDA over PLGA nanoparticles is not referred to in the introduction  
  • other diosgenin studies in nanoparticles are missing in the introduction
  • there is no explanation why these ratios of PGMD are used for the formulations
  • the complete polymer characterization is missing! such as molar mass, characterization, purification,...
  • how are the nanoparticles characterized in terms of "surface charge density" as mentioned in the manuscript? The zeta potential does not give this number!
  • what is the reaction with perchloric acid and diosgenin? what is the resulting colored product? Did you make blank studies with only polymer and no drug? what is the detection wavelength?
  • How can you be sure that no nanoparticles are left in the supernatant after 10 min centrifugation at 12 krpm that might disturb the drug release kinetics?
  • Axes in Figure 1 are not readable
  • what do you mean with "optimized nanoparticles" in 3.3?
  • A PDI of 0.34 is not at all good for an optimized prpeparation as outlined in 3.3.
  • chapter 3.4. should be introduced much earlier  
  • there is no real difference to be discussed in EE% in chapter 3.5., the values are the same within the errors
  • chapter 3.4.: there is no sustained release after 24 h because there is no change in drug release after 24 h
  • The difference in EE% inTable 3 are not significant and within the error-. 
  • do you have any data about the degradation of the nanoparticles since there is no 100% release? The nanoparticles should be degraded or show signs of degradation
  • Any control of the pure unloaded nanoparticles in the in vitro tests are missing
  • The images showing apoptosis (Figure 8+9) are not convincing. Image analysis is needed to draw any conclusion. Scale bars are missing.
  • what is the reason for a better performance of the drug loaded nanoparticles versus free drug?
  • what is the control in Figure 10A and B; no particles or drug? 

minor points: 

  • DLS data give always size distributions of the nanoparticles; it is therefore not necessary to mention the number after the comma when specifying the z-average diameter  
  • last sentence of chapter 3.4: do you mean after the synthesis?

Author Response

Response to comments of reviewers on the manuscript entitled, “Diosgenin loaded polymeric nanoparticles with potential anticancer efficacy”.

We would like to thank the editors for reviewing our manuscript. We have carefully gone through the comments/suggestions and made the necessary changes in the revised manuscript. We sincere hope that the revision has brought the manuscript to the level of reviewer’s satisfaction. Please find below our point-by-point reply to the comments and the modifications incorporated in the manuscript thereafter has been highlighted in yellow.

Response to the comments

Reviewer: 1

Comment 1- the benfit of DDA over PLGA nanoparticles is not referred to in the introduction.

Reply: The benefit of PGMD over PLGA has been mentioned in the introduction section of the manuscript Lines 89-100.

Comment 2- other diosgenin studies in nanoparticles are missing in the introduction.

Reply: As suggested, the studies related to diosgenin encapsulated nanoparticles have been incorporated in the revised manuscript in the introduction section Lines 101-114.

Comment 3- there is no explanation why these ratios of PGMD are used for the formulations.

Reply: Our group in one of the previous studies have modified the PGD by adding malic acid and adjusting ratio of malic acid to dodecanedioate (DDA) during the synthesis process to make it less hydrophobic. To alter the hydrophobicity of the PGMD polymer, several variants of the polymer were synthesized with different ratios of malic acid to DDA [1]. Further, amongst all the variants two ratios of 6:4 and 7:3 used for the preparation of nanoparticles and the in vitro drug delivery efficiency gave best results. The explanation behind the use of these two ratios has been inserted in the revised manuscript under section 3 Results and discussion, lines 238-251. 

Comment 4- the complete polymer characterization is missing! such as molar mass, characterization, purification,...

Reply: We would like to mention that the molecular weight of PGMD polymer as measured through GPC column is around 3000 Da. The glass transition temperature of the polymer was 42.2°C as determined through DSC [1]. This information has been inserted in the revised manuscript under section 3 Results and discussion, lines 238-251. This information has been reported by our group in one of the earlier publication, hence, have not mentioned the characterization studies in this manuscript [1].

Comment 5- how are the nanoparticles characterized in terms of "surface charge density" as mentioned in the manuscript? The zeta potential does not give this number!

Reply: The zeta potential is basically the value of the electrostatic potential of the surface of hydrodynamic shear, i.e., the surface where particle interact with one another or with surfaces. The apparent magnitude of zeta potential indicates the dispersion stability against aggregation. As pointed out by the reviewer, the statement has been modified in the revised manuscript under section 2.4 lines 165-166.

Comment 6- what is the reaction with perchloric acid and diosgenin? what is the resulting colored product? Did you make blank studies with only polymer and no drug? what is the detection wavelength?

Reply: Steroidal saponins such as diosgenin possess hydroxyl group at the 3beta position and a double bond at 5-6 position of the carbon ring. These reacts with perchloric acid to give yellow coloured chromogens with the absorbance at 410nm. The entire structure of the coloured product is not been elucidated [2,3]. We would like to mention that in the initial experiment, the U.V. absorption of the PGMD polymer was measured and no specific absorbance was found in the spectral scan (200-800nm). So, during the absorbance of drug polymer do not show any interference in the particular wavelength range.

Comment 7- How can you be sure that no nanoparticles are left in the supernatant after 10 min centrifugation at 12 krpm that might disturb the drug release kinetics?

Reply: The statement has been revised to “At different time intervals, the aliquots were taken out and centrifuged at 12,000 rpm for 20min” lines 194-195. Initially the centrifugation speed was optimized for pelleting the nanoparticles. It was observed that particles when suspended in MilliQ water at 12000 rpm for 20 min, the supernatant showed no specific absorbance of drug and drug loaded nanoparticles. When placed in acidic medium, hydrolysis of ester bond resulted in the release of drug molecules from the nanoparticles, which was further estimated through drug release kinetic studies. 

Comment 8- Axes in Figure 1 are not readable

Reply: As pointed out by the reviewer figure 1 has been revised with better axes labelling. 

Comment 9- what do you mean with "optimized nanoparticles" in 3.3? A PDI of 0.34 is not at all good for an optimized prpeparation as outlined in 3.3.

Reply: As mentioned in section 3.3, “optimized nanoparticles” were referred as the optimum conditions required for the synthesis of nanoparticles with particle size <200nm and homogenous size distribution. Though the PDI values of 0.2 and below are most commonly acceptable in practice, but for drug delivery applications a PDI of 0.35 and below is considered to be acceptable and indicates monomodal distribution of particles [4,5]. Further, the PDI of 0.34 was predicted by BBD which upon experimental validation found to be 0.152 which is quite acceptable.

Comment 10- chapter 3.4. should be introduced much earlier. there is no real difference to be discussed in EE% in chapter 3.5., the values are the same within the errors.

Reply: The main focus of the present study was to explore the potential of novel polymer PGMD for its delivery efficiency and compatibility of the anticancer drug. With regard to this, initially BBD approach was used to obtain the optimum condition for synthesizing diosgenin loaded PGMD nanoparticles. Further particles were synthesized under optimum condition and were characterized for their particle size and distribution, zeta potential, encapsulation and loading efficiency and in vitro studies. Therefore, the section 3.4 for characterisation was introduced after the synthesis part. We agree with the reviewer, that there is no significant difference in the encapsulation efficiency between the polymeric nanoparticles. The statement has been revised in the revised manuscript lines 382-383.     

Comment 11- chapter 3.4.: there is no sustained release after 24 h because there is no change in drug release after 24 h.

Reply: Under 3.6 “In vitro drug release analysis” section the nanoparticles were assessed for their release pattern in different pH buffer over a defined time period. It was evident from the results that a biphasic release pattern was observed from both the variants of polymer. The burst release for diosgenin could be associated with the release of non-encapsulated or loosely bounded drug in the polymer matrix, which was easily accessible on hydration of the nanoparticles [6]. The release was slower for the next 24h with upto 70% release (for both the PGMD variants i.e. 7:3, 6:4), due to the hydrophobic interaction of polymer-drug within the nanoparticle. This resists the fast diffusion of the drug from the particle and maintained its sustained release for long hours. Similar results were obtained when doxorubicin loaded PGMD nanoparticles were developed. It was observed that ~ 20% doxorubicin was released in 29 days from the nanoparticles [7]. Therefore, the results dictate a slower and prolonged drug release from the nanoparticles.   

Comment 12- The difference in EE% in Table3 are not significant and within the error-

Reply: We agree with the suggestion/comment of the reviewer, that the encapsulation efficiency between the two nanoformulations was not significant and the statement in the manuscript has been revised, lines 382-383.    

Comment 13- do you have any data about the degradation of the nanoparticles since there is no 100% release? The nanoparticles should be degraded or show signs of degradation

Reply: We have not conducted polymeric degradation studies in the present work. It has been reported in literature that under acidic environmental conditions, hydrolysis of ester bond takes place. Because of this, biphasic pattern could be observed from the nanoparticles, burst release of drug during the initial release studies and further slower release of the drug. The results were in agreement with the previous studies [7]. Also, it was shown that ~20% hydrophobic doxorubicin was released from the PGMD nanoparticles after 29 days under acidic environment. So, it can be speculated that the PGMD nanoparticles showed slow and sustained release of drug from the nanoparticles.

Comment 14- Any control of the pure unloaded nanoparticles in the in vitro tests are missing

Reply: We would like to mention here that void PGMD nanoparticles did not showed cytotoxic effect in the cells. It has already been evident in the previous studies of our group that the void PGMD nanoparticles do not exhibit toxic effect in the MES-SA and Dx5 cells [8]. Hence, for in vitro cell toxicity studies herein, cells seeded in complete medium were taken as negative control.

Comment 15- The images showing apoptosis (Figure 8+9) are not convincing. Image analysis is needed to draw any conclusion. Scale bars are missing.

Reply: As suggested by the reviewer, the images have been corrected and scale bar has been added in the images. AO/EtBr is a qualitative method used to determine live, early, late apoptotic and necrotic cancer cells by observing nuclear changes in the cells [9]. The presence of yellow and red fluorescence in the images with chromatin fragmentation after the treatment with free diosgenin, PGMD-DG 6:4, PGMD-DG 7:3 NPs suggests that PGMD-DG NPs (both variants) largely induce apoptosis in A549 cancer cells. The images have been quantitated using Image J and the quantitative data presented in form of bar graphs in figure 8. Moreover, to observe chromatin fragmentation, multinucleation, cytoplasmic vacuolation, nuclear swelling and late apoptosis, DAPI staining in A549 cells was performed. The presence of blue fluorescence with chromatin condensation after the treatment depicts that PGMD nanoparticle encapsulated diosgenin induce apoptosis largely in the cells as compared to free drug. The results were in accordance with literature [10].

Comment 16- what is the reason for a better performance of the drug loaded nanoparticles versus free drug?

Reply: The enhancement of cytotoxic activity of the drug loaded nanoparticles than the free drug can be due to many reasons. According to the literature, nanoparticles gets adsorb onto the cell membrane, leading to an increase in drug concentration near the cell surface, thus resulting in the generation of concentration gradient that would favour the drug influx into the cell. Secondly, tumor cells (which possess enhanced endocytic activity) could internalise polymeric nanoparticles allowing the drug to be release inside the cells, thus contributing to an increase in the drug concentration near its site of action [11]. Additionally, it is also possible that this could be the result of high payload of diosgenin in the nanoparticles. This could result in substantial intracellular delivery even if only a small number of nanoparticles were able to enter the cells [12].   

Comment 17- what is the control in Figure 10A and B; no particles or drug?

Reply: The control in figure 10 A and 10 B which in the revised manuscript are figure 9A and 9B,  was the cells seeded in complete medium without the treatment of drug and nanoparticles.

Minor points

Comment 18- DLS data give always size distributions of the nanoparticles; it is therefore not necessary to mention the number after the comma when specifying the z-average diameter

Reply: We agree with the reviewer that DLS data gives size distributions of the nanoparticles.

Comment 19- last sentence of chapter 3.4: do you mean after the synthesis?

Reply: As mentioned, the surfactant present in the solution increases the surface tension resulting in the formation of smaller nanoparticles. In addition to this, it also avoids the aggregation of nanoparticles during the entire preparation process not after the synthesis.

  1. Lei, T.; Manchanda, R.; Huang, Y.-C.; Fernandez-Fernandez, A.; Bunetska, K.; Milera, A.; Sarmiento, A.; McGoron, A.J. Near-infrared imaging loaded polymeric nanoparticles: in vitro and in vivo studies. In Proceedings of Reporters, Markers, Dyes, Nanoparticles, and Molecular Probes for Biomedical Applications V; p. 859607.
  2. Baccou, J.; Lambert, F.; Sauvaire, Y. Spectrophotometric method for the determination of total steroidal sapogenin. Analyst 1977, 102, 458-465.
  3. Zhang, F.; Shen, B.; Jiang, W.; Yuan, H.; Zhou, H. Hydrolysis extraction of diosgenin from Dioscorea nipponica Makino by sulfonated magnetic solid composites. Journal of Nanoparticle Research 2019, 21, 269.
  4. Danaei, M.; Dehghankhold, M.; Ataei, S.; Hasanzadeh Davarani, F.; Javanmard, R.; Dokhani, A.; Khorasani, S.; Mozafari, M. Impact of particle size and polydispersity index on the clinical applications of lipidic nanocarrier systems. Pharmaceutics 2018, 10, 57.
  5. Tatielle do Nascimento, E.R.-J. Preparation of Microparticles of Polycaprolactone Containing Piroxicam to use in Chronic Inflammatory Diseases. The Pharmaceutical and Chemical Journal 2017, 4, 154-163.
  6. Wang, J.; Wang, B.M.; Schwendeman, S.P. Characterization of the initial burst release of a model peptide from poly (D, L-lactide-co-glycolide) microspheres. Journal of controlled release 2002, 82, 289-307.
  7. Lei, T.; Manchanda, R.; Fernandez-Fernandez, A.; Huang, Y.-C.; Wright, D.; McGoron, A.J. Thermal and pH sensitive multifunctional polymer nanoparticles for cancer imaging and therapy. RSC advances 2014, 4, 17959-17968.
  8. Lei, T.; Fernandez-Fernandez, A.; Manchanda, R.; Huang, Y.-C.; McGoron, A.J. Near-infrared dye loaded polymeric nanoparticles for cancer imaging and therapy and cellular response after laser-induced heating. Beilstein journal of nanotechnology 2014, 5, 313-322.
  9. Pillai, J.J.; Thulasidasan, A.K.T.; Anto, R.J.; Devika, N.C.; Ashwanikumar, N.; Kumar, G.V. Curcumin entrapped folic acid conjugated PLGA–PEG nanoparticles exhibit enhanced anticancer activity by site specific delivery. RSC advances 2015, 5, 25518-25524.
  10. Bhatnagar, P.; Kumari, M.; Pahuja, R.; Pant, A.; Shukla, Y.; Kumar, P.; Gupta, K. Hyaluronic acid-grafted PLGA nanoparticles for the sustained delivery of berberine chloride for an efficient suppression of Ehrlich ascites tumors. Drug delivery and translational research 2018, 8, 565-579.
  11. Fonseca, C.; Simoes, S.; Gaspar, R. Paclitaxel-loaded PLGA nanoparticles: preparation, physicochemical characterization and in vitro anti-tumoral activity. Journal of controlled release 2002, 83, 273-286.
  12. Betancourt, T.; Brown, B.; Brannon-Peppas, L. Doxorubicin-loaded PLGA nanoparticles by nanoprecipitation: preparation, characterization and in vitro evaluation. 2007.

Reviewer 2 Report

The manuscript from Sharma et al, is well written and easily readable. However, there are both major and minor revisions that are needed.

Concerning BBD, the criteria for the selection of the formulations are not clear. It seems that the selection was based on the best correlation between theoretical end experimental. Actually, the model is not giving clear information. The selection of the formulation should take into account the correlation, but mainly the features of the obtained nanoparticles. This latter aspect should be more emphasised.

Concerning the results, they must provide polymer features, at least Mw, polydispersity, and composition as well as synthesis yield. Otherwise, the synthesis could be removed from the methods and the characteristics of the polymer provided in the list of materials, instead.

Figure 7 is useless. The release profiles are already shown in figure 6 and the correlation with releasing models is more correctly reported with the data in table 4.

Concerning the biological evaluations, the reported micrographs do not support the discussion. The authors provide a qualitative investigation for apoptosis which is not clearly revealed by the micrographs. Higher magnification must be provided to correctly interpret the results. Additionally, figure 8 is eliminable since is not providing additional information with respect to figure 9.

Mainor revisions :

The term synthesis is used improperly all over the document in place of preparation of NP. The polymer is synthesized, the particles are obtained from a preparation. There is no covalent (synthetic) bond forming during particle formation.

Line 36: insert reference for the consulted WHO report.

Line 104: PF-68 should be pluronic F68

Line 172: “entrapped” is to be changed with “released

Author Response

Response to comments of reviewers on the manuscript entitled, “Diosgenin loaded polymeric nanoparticles with potential anticancer efficacy”.

We would like to thank the editors for reviewing our manuscript. We have carefully gone through the comments/suggestions and made the necessary changes in the revised manuscript. We sincere hope that the revision has brought the manuscript to the level of reviewer’s satisfaction. Please find below our point-by-point reply to the comments and the modifications incorporated in the manuscript thereafter has been highlighted in yellow.

Reviewer: 2

Comment 1- Concerning BBD, the criteria for the selection of the formulations are not clear. It seems that the selection was based on the best correlation between theoretical end experimental. Actually, the model is not giving clear information. The selection of the formulation should take into account the correlation, but mainly the features of the obtained nanoparticles. This latter aspect should be more emphasised.

Reply:  We agree with the reviewer that the selection of formulation should be based on the parameters that dictate the nanoparticles size and polydispersity. We would like to mention here that the parameters that could dictate the size of the particles prepared here i.e. the concentration of PGMD (mg), diosgenin (mg) and PF-68 (%), have been selected on the basis of some information available with us from some of our earlier publication on PGMD nanoparticles mediated drug delivery studies and in the literature [1] [2]. The BBD suggested formulations were prepared and later the data obtained was submitted for generation of the response curves.    

Comment 2- Concerning the results, they must provide polymer features, at least Mw, polydispersity, and composition as well as synthesis yield. Otherwise, the synthesis could be removed from the methods and the characteristics of the polymer provided in the list of materials, instead.

Reply: We would like to mention that the molecular weight of PGMD polymer as measured through GPC column is around 3000 Da. The glass transition temperature of the polymer was 42.2°C as determined through DSC [3]. The characterisation of the polymer was shown by our group in the previous studies. Therefore, the same has not been included in the present study. However, this information has been inserted in the revised manuscript under section 3 Results and discussion, lines 238-251. 

Comment 3- Figure 7 is useless. The release profiles are already shown in figure 6 and the correlation with releasing models is more correctly reported with the data in table 4.

Reply: We would like to mention that figure 6 depicts the drug release studies at different pH under various time intervals. In order to make a clear explanation, the releasing kinetic models for both the nanoformulations were shown in figure 7. Table 4 shows the correlation values for the different models. However, respecting the suggestion of the reviewer we have removed figure 7 from the revised manuscript.

Comment 4- Concerning the biological evaluations, the reported micrographs do not support the discussion. The authors provide a qualitative investigation for apoptosis which is not clearly revealed by the micrographs. Higher magnification must be provided to correctly interpret the results. Additionally, figure 8 is eliminable since is not providing additional information with respect to figure 9.

Reply: In the present study, morphological changes in the cell after the administration of nanoparticles were analysed microscopically through acridine orange-ethidium bromide (AO/EtBr) dual staining, and DAPI staining. AO/EtBr is a qualitative method used to determine live, early, late apoptotic and necrotic cancer cells by observing nuclear changes in the cells. AO penetrates the intact membrane of usual and early apoptotic cell and binds to DNA, which fluorescese uniform green in normal cells and as patches in early apoptotic cells due to chromatin condensation. While, EtBr is only penetrable in the compromised membrane of late apoptosis and necrotic cell, where it fluoresces as orange patch by its binding to fragmented DNA or apoptotic body in late apoptotic or necrotic cells[4]. The presence of yellow and red fluorescence in the images with chromatin fragmentation after the treatment with free diosgenin, PGMD-DG 6:4, PGMD-DG 7:3 NPs suggests that PGMD-DG NPs (both variants) largely induce apoptosis in A549 cancer cells. The images have been quantitated using Image J and the quantitative data presented in form of bar graphs in figure 8.

Moreover, to observe chromatin fragmentation, multinucleation, cytoplasmic vacuolation, nuclear swelling and late apoptosis, DAPI staining in A549 cells was performed. The presence of blue fluorescence with chromatin condensation after the treatment depicts that PGMD nanoparticle encapsulated diosgenin induce apoptosis largely in the cells as compared to free drug. The results were in accordance with literature [5]. Recently, Li et al., showed that CURSN38 NPs showed enhanced level of apoptosis as compared to drug alone in Hela cells as determined through AO/EtBr and DAPI staining [6]. As suggested by the reviewer, modified images with high magnification have been added in the manuscript.           

Minor points

Comment 5- The term synthesis is used improperly all over the document in place of preparation of NP. The polymer is synthesized, the particles are obtained from a preparation. There is no covalent (synthetic) bond forming during particle formation.

Reply: The word synthesis has been replaced in the manuscript. The changes have been highlighted in yellow colour.

Comment 6- Line 36: insert reference for the consulted WHO report.

Line 104: PF-68 should be pluronic F68

Line 172: “entrapped” is to be changed with “released

Reply: As mentioned, the statements have been revised in the manuscript with change in highlighted in yellow.

References

  1. Lei, T.; Manchanda, R.; Fernandez-Fernandez, A.; Huang, Y.-C.; Wright, D.; McGoron, A.J. Thermal and pH sensitive multifunctional polymer nanoparticles for cancer imaging and therapy. RSC advances 2014, 4, 17959-17968.
  2. Lei, T.; Fernandez-Fernandez, A.; Manchanda, R.; Huang, Y.-C.; McGoron, A.J. Near-infrared dye loaded polymeric nanoparticles for cancer imaging and therapy and cellular response after laser-induced heating. Beilstein journal of nanotechnology 2014, 5, 313-322.
  3. Lei, T.; Manchanda, R.; Huang, Y.-C.; Fernandez-Fernandez, A.; Bunetska, K.; Milera, A.; Sarmiento, A.; McGoron, A.J. Near-infrared imaging loaded polymeric nanoparticles: in vitro and in vivo studies. In Proceedings of Reporters, Markers, Dyes, Nanoparticles, and Molecular Probes for Biomedical Applications V; p. 859607.
  4. Pillai, J.J.; Thulasidasan, A.K.T.; Anto, R.J.; Devika, N.C.; Ashwanikumar, N.; Kumar, G.V. Curcumin entrapped folic acid conjugated PLGA–PEG nanoparticles exhibit enhanced anticancer activity by site specific delivery. RSC advances 2015, 5, 25518-25524.
  5. Bhatnagar, P.; Kumari, M.; Pahuja, R.; Pant, A.; Shukla, Y.; Kumar, P.; Gupta, K. Hyaluronic acid-grafted PLGA nanoparticles for the sustained delivery of berberine chloride for an efficient suppression of Ehrlich ascites tumors. Drug delivery and translational research 2018, 8, 565-579.
  6. Li, X.; Gao, Y. Synergistically fabricated polymeric nanoparticles featuring dual drug delivery system to enhance the nursing care of cervical cancer. Process Biochemistry 2020, 98, 254-261.

Round 2

Reviewer 1 Report

The authors revised the manuscript. Within the revision, the style and language needs to be improved as well as typos be removed. And there are still some points that need to be corrected. 

Comment 1: The manuscript includes now few points about PLGA however these are all advantages over other polymers (better stability, good encapsulation of drugs) and no reason to use PGMD. This point is not addressed.

Comment 4: the characterization data need to be included in the experimental section since the authors also mention that there are slight modifications in the protocol of the literature, the real data of the used polymers need to be included in the paper such as GPC data (solvent, calibration, dispersity,...)

Comment 7: Please proved the data (spectra) here.

Comment 9: A PDI of 0.34 does not mean monomodal distribution. The DLS data are not provided, but a PDI above 0.2 means that there is either a broad distribution or smaller and larger fractions. This is far away from optimized formulations.     

Comment 11: I still do not see any release after 24 h. The authors should add any data about release over 29 days as they mentioned. In this paper, the particles show a burst release but then they seem to be stable and do not release the residual drug.

Comment 13: Degradation studies are essential in case a degradable polymer is proposed! Burst release does not consequently refer to degradation because in case of degradation, the particles would not exist anymore and you would have a 100% release.

Comment 14: The reference to the control should be mentioned then.

Comment 15: The meaning of the errors need to be mentioned in the caption.

Comments 16: such explanations may be added to the manuscript.

Comment 17: so consequently the colour code of the control makes no sense. Does it mean repetitions? Should be the same for the statistics as in the other experiments.  

Author Response

Response to comments of reviewers on the manuscript entitled, “Diosgenin loaded polymeric nanoparticles with potential anticancer efficacy”.

We would like to thank the editors for again reviewing our manuscript. We have carefully gone through the comments/suggestions and made the necessary changes in the revised manuscript. We sincere hope that the revision has brought the manuscript to the level of reviewer’s satisfaction. Please find below our point-by-point reply to the comments and the modifications incorporated in the manuscript thereafter has been highlighted in green.

Reviewer-1

Comment 1: The manuscript includes now few points about PLGA however these are all advantages over other polymers (better stability, good encapsulation of drugs) and no reason to use PGMD. This point is not addressed.

Reply: As per the suggestion, the description for using PGMD has been mentioned from the line number 95-106.

Comment 4: the characterization data need to be included in the experimental section since the authors also mention that there are slight modifications in the protocol of the literature, the real data of the used polymers need to be included in the paper such as GPC data (solvent, calibration, dispersity,...)

Reply: The polymer was characterized by NMR and FTIR and the data has been submitted as supplementary data in figure 5 and 6. Further, we would like to bring to the notice of the reviewers that our University including our lab was closed since March 21, 2020 due to Corona pandemic and normal functioning along with research activities are seriously hampered, we are working with almost one-third workforce. Though, we welcome the suggestion of the reviewer to present GPC data, it was not possible for us to perform the GPC analysis at this time. With this present situation and limited resources we have tried our sincere efforts to improve the manuscript and have incorporated the 1HNMR and FTIR. We hope that the reviewer will understand our situation and exempt from GPC studies.

Comment 7: Please proved the data (spectra) here.

Reply:  As mentioned earlier, after centrifugation at 12000 rpm for 20 min, not much nanoparticles are present in the nanoparticles. The spectrum of supernatant showed no peculiar peak at 410nm as observed in the diosgenin spectrum.

Comment 9: A PDI of 0.34 does not mean monomodal distribution. The DLS data are not provided, but a PDI above 0.2 means that there is either a broad distribution or smaller and larger fractions. This is far away from optimized formulations.

Reply: As pointed out now we have included the PDI values along with the nanoparticles size in the manuscript in the lines 364-367 and the DLS spectra has been submitted as supplementary data as figures1-4. As can be seen from the DLS data the experimental value of PDI for the nanoparticles obtained were 0111, 0.152 and 0.137, 0.270 which are acceptable and dictates unimodal distribution of particles. Though the experimental value deviates from theoretical approach, but still the particles obtained were in small size with uniform size distribution.

Comment 11: I still do not see any release after 24 h. The authors should add any data about release over 29 days as they mentioned. In this paper, the particles show a burst release but then they seem to be stable and do not release the residual drug.

Reply: We strongly agree with the reviewer’s point of performing the release assay for 29 days due to its stable and slow release profile. However, the study was conducted only for 48 hrs, as it was evident from previous reports that the nanoformulations showed slow release of drug. Regardless, we take into account your valuable suggestion for future studies.

Comment 13: Degradation studies are essential in case a degradable polymer is proposed! Burst release does not consequently refer to degradation because in case of degradation, the particles would not exist anymore and you would have a 100% release.

Reply: We agree to the reviewer’s point, unfortunately, we haven’t conducted the experiment for degradation and regret for not being able to add the result. As previously mentioned, our  results were in concordance with the study performed previously and that the release showed biphasic release pattern, with initial burst release, followed by slow release of drug (1).

Comment 14: The reference to the control should be mentioned then.

Reply: As suggested, the reference has been added in the manuscript (line 495-496).

Comment 15: The meaning of the errors need to be mentioned in the caption.

Reply: As suggested, the p value has been added in the caption (lines 488-489).

Comments 16: such explanations may be added to the manuscript.

Reply: As suggested by the reviewer, the explanations have discussed in the manuscript and highlighted with yellow (line 504 to 513).

Comment 17: so consequently the colour code of the control makes no sense. Does it mean repetitions? Should be the same for the statistics as in the other experiments.

Reply: The figure 9 has been corrected with single control bar including standard deviations.

References

  1. Lei T, Manchanda R, Fernandez-Fernandez A, Huang Y-C, Wright D, McGoron AJ. Thermal and pH sensitive multifunctional polymer nanoparticles for cancer imaging and therapy. RSC advances. 2014;4(34):17959-68.

Reviewer 2 Report

The manuscript was amended according to previous suggestions. In the conclusions paragraph, the term “synthesis” for nanoparticles preparation is still used and should be substituted, anyway.

Author Response

Response to comments of reviewers on the manuscript entitled, “Diosgenin loaded polymeric nanoparticles with potential anticancer efficacy”.

We would like to thank the editors for again reviewing our manuscript. We have carefully gone through the comments/suggestions and made the necessary changes in the revised manuscript. We sincere hope that the revision has brought the manuscript to the level of reviewer’s satisfaction. Please find below our point-by-point reply to the comments and the modifications incorporated in the manuscript thereafter has been highlighted in green.

Reviewer 2

The manuscript was amended according to previous suggestions. In the conclusions paragraph, the term “synthesis” for nanoparticles preparation is still used and should be substituted, anyway.

Reply: As suggested, the sentence has been revised and highlighted in line 522.
